# Comparative Genomic Analysis of the Poaceae Cytokinin Response Regulator *RRB* Gene Family and Functional Characterization of *OsRRB5* in Drought Stress Tolerance in Rice

**DOI:** 10.3390/ijms26051954

**Published:** 2025-02-24

**Authors:** Rujia Chen, Qianfeng Huang, Yanan Xu, Zhichao Wang, Nian Li, Yue Lu, Tianyun Tao, Yu Hua, Gaobo Wang, Shuting Wang, Hanyao Wang, Yong Zhou, Yang Xu, Pengcheng Li, Chenwu Xu, Zefeng Yang

**Affiliations:** 1Jiangsu Key Laboratory of Crop Genomics and Molecular Breeding/Zhongshan Biological Breeding Laboratory/Key Laboratory of Plant Functional Genomics of the Ministry of Education/Jiangsu Key Laboratory of Crop Genetics and Physiology, Agricultural College of Yangzhou University, Yangzhou 225009, China; rjchen@yzu.edu.cn (R.C.); 19826288587@163.com (Q.H.); 13813069377@163.com (Y.X.); dx120240161@stu.yzu.edu.cn (Z.W.); l2512632@126.com (N.L.); luyue@yzu.edu.cn (Y.L.); tytao@yzu.edu.cn (T.T.); dx120230146@stu.yzu.edu.cn (Y.H.); ang3553561877@163.com (G.W.); wst13270565587@126.com (S.W.); xln20000301@126.com (H.W.); zhouyong@yzu.edu.cn (Y.Z.); yangx@yzu.edu.cn (Y.X.); pcli@yzu.edu.cn (P.L.); 2Jiangsu Co-Innovation Center for Modern Production Technology of Grain Crops, Yangzhou University, Yangzhou 225009, China; 3Joint International Research Laboratory of Agriculture and Agri-Product Safety of the Ministry of Education, Yangzhou University, Yangzhou 225009, China

**Keywords:** cytokinin response regulator, type B *RR* gene, phylogenetic and synteny analysis, gene expression pattern, haplotype variation analysis

## Abstract

The cytokinin (CK) type B response regulator (*RRB*) gene is involved in the CK signaling pathway and performs a key function for mediating reactions to amounts of abiotic stresses. Nevertheless, the *RRB* gene family remains to be characterized in Poaceae (also known as Gramineae or grasses). Here, we performed a comprehensive analysis encompassing phylogenetic relationships, evolutionary pressures, and expression patterns of the *RRB* gene family in six Poaceae species, including rice, *Panicum*, *Sorghum*, *Setaria*, maize, and wheat. Phylogenetic tree and syntenic analyses revealed that the *RRB* genes were divided into seven orthologous gene clusters (OGCs), indicating that the common ancestor of these Poaceae species possessed at least seven *RRB* genes. Further analysis revealed that the evolution of the Poaceae *RRB* gene family was primarily driven by purifying selection. The expression pattern of rice *OsRRB* toward phytohormonal and abiotic stresses was also investigated. The findings revealed that several phytohormones, including cytokinin (CK), abscisic acid (ABA), and methyl jasmonate (MeJA), as well as abiotic factors such as drought and cold, significantly increased the expression levels of these genes. Importantly, haplotype analysis identified four crucial variation sites within the *OsRRB5* genomic regions that may contribute to drought resistance in rice. Our findings lay the groundwork for further elucidating the biological function of *OsRRB* genes and provide a promising new target for developing stress-resistant rice varieties.

## 1. Introduction

Cytokinin (CK) is a crucial class of important phytohormones that play significant roles in multiple biological processes. They are reported to be extensively involved in plant growth and development, including cell division, shoot initiation, leaf and vascular tissue production, root architecture, photosynthetic performance, light responses, and nutrient uptake, and in reproductive processes, including gametophyte and embryo formation [1,2]. In addition, CKs are known to respond to both abiotic and biotic stresses, which include cold, salinity, dryness, and osmotic adaptation, as well as plant–microbe interactions [3,4,5]. In plants, CK is produced through a multistep phosphorelay (MSP) mechanism that resembles the two-component systems (TCSs) used by bacteria [6,7]. Characterization of MSP indicates that the CK signaling pathway involves three key components: the CK receptors histidine-aspartate kinases (HKs), the histidine-containing phosphotransfer (HPts) proteins, and the response regulators (RRs), all of which have been identified across various plant species [8], including *Arabidopsis* [9], tomato [10], rice [11,12,13,14], maize [15], and wheat [16].

The response regulator (*RR*) family serve as the final phosphoacceptor molecules in the His–Asp phosphorelay. This family can be categorized into four distinct subgroups, the type A response regulators (RRA), type B response regulators (RRB), type C response regulators (RRC), and clock-related response regulators (PRR), based on the amino acid composition [9,17,18]. As a primary response factor, *RRA* gene expression is rapidly induced by CKs [19]. In contrast, phosphorylated *RRB* genes function as transcriptional activators, triggering the swift induction of CK-related target genes, including a subset of *RRA* genes [20]. Notably, *RRB* genes not only regulate CK signaling pathways, but also are essential for mediating responses to various abiotic stresses [3,21,22,23]. For instance, the *Arabidopsis RRB* triple mutant *arr1/arr10/arr12* showed significantly enhanced drought resistance compared with wild-type plants (WT) as evidenced by higher relative moisture content and survival under desiccating conditions. It suggests that *ARR1*, *ARR10*, and *ARR12* are negative regulators of drought stress response for *Arabidopsis* [23]. Additionally, in shoots, *Arabidopsis ARR1* and *ARR12* genes regulate sodium levels by fine-tuning the expression of gene-encoding high-affinity K+ transporter 1;1 (*AtHKT1;1*), thereby significantly impacting salt tolerance [3]. *ARR18*, another *RRB* gene, acts in enhancing osmotic stress resistance for *Arabidopsis* seeds by promoting the expression of the proline dehydrogenase 1 gene (*PDH1*) [24]. Nevertheless, the role of *RRB* genes in Poaceae, especially in rice, remain not totally explored.

Poaceae (also known as Gramineae or grasses) is a large and widely distributed group of monocot flowering plants. It is one of the most successful plant groups, with significant economic, ecological, and evolutionary importance. Although genome-wide analyses have identified *RR* gene family members in certain plant species [25,26,27,28], a more comprehensive understanding on the evolutionary patterns of *RRB* genes in Poaceae, particularly regarding selective pattern profiles, remains to be conducted. In this research, a thorough analysis was performed, including the phylogenetic relationships, evolutionary pressures, and expression patterns in response to phytohormones and environmental stressors of the *RRB* gene family in six representative species of Poaceae. Furthermore, we investigated the haplotype variations in the rice *OsRRB5* gene. Our findings not only lay the foundation for further elucidating the role and function of *OsRRB* genes, but also offer new perspectives for its potential application in rice stress-resistance breeding.

## 2. Results

### 2.1. Identification and Characterization of the RRB Genes in Poaceae

Based on BLASTP searches against protein sequences and Batch CD searches for the conserved REC_typeB_ARR-like domain (cd17584), a total of 60 *RRB* genes were identified across 6 representative species of Poaceae (Table 1). Specifically, six genes identified in *Oryza sativa* are designated as *OsRRB1* to *OsRRB6* in order of their chromosomal locations. The corresponding names (*OsRR21* to *OsRR26*) from the previous report [9] are also provided in Table 1. Similarly, 7 genes in *Panicum hallii* are designated as *PhRRB1* to *PhRRB7*, 11 genes in *Sorghum bicolor* are designated as *SbRRB1* to *SbRRB11*, 9 genes in *Setaria italica* are designated as *SiRRB1* to *SiRRB*9, 9 genes in *Zea mays* are designated as *ZmRRB1* to *ZmRRB9*, and 18 genes in *Triticum aestivum* are designated as *TaRRB1* to *TaRRB18*, respectively (Table 1). In addition, the characterization of RRB proteins was analyzed, encompassing the length of amino acid sequence, molecular weight, theoretical isoelectric point (pI), instability index, aliphatic index, grand average of hydration, and predicted subcellular localization (Table 1). The amino acid sequence lengths of RRB proteins varied considerably, ranging from 258 (*TaRRB13*) to 753 (*PhRRB4*) amino acids, with theoretical molecular weights ranging from 29,667.3 to 81,469.49 Da. Analysis of the isoelectric point revealed that the pI values of the 60 RRB proteins ranged from 5.04 to 7.18, with an average of approximately 5.96, indicating that they might function in weakly acidic cellular environments. Among the 60 RRB proteins, only 11 exhibited an instability index < 40, implying that the majority of these proteins were instable. The grand average of hydropathicity values of the 60 RRB proteins is less than 0, suggesting that they are hydrophilic proteins. Predicted subcellular localization revealed that PhRRB4 and TaRRB8 are located in the chloroplast, TaRRB13 in the cytoplasm, SbRRB11 in the vacuole, and ZmRRB9 in the peroxisome, while the others are located in the nucleus. These findings indicate that the 60 RRB proteins exhibit diverse basic physical and chemical properties, reflecting the numerous biological functions they are involved in.

### 2.2. Phylogenetic Relationships of the RRB Genes in Poaceae

To elucidate the phylogenetic relationships of Poaceae *RRB* genes, a combined phylogenetic tree was generated based on the 60 aligned RRB protein sequences, employing both the maximum likelihood and neighbor-joining methods (Figure 1A). In the phylogenetic analysis, the Poaceae *RRB* genes were categorized into seven orthologous gene clusters (OGCs), implying that the common ancestor of these Poaceae species likely possessed at least seven *RRB* genes. However, we observed variations in gene numbers among these species across the seven OGCs, likely resulting from gene duplication and/or loss. The rice genome contained six *RRB* genes but lacked one gene in OGC7, suggesting a single gene loss event. The *Panicum* genome possessed seven *RRB* genes and exhibited one gene duplication event and one loss event. The *Sorghum* genome contained 11 *RRB* genes, and exhibited a minimum of 3 duplicate events and 1 loss event. In contrast, the *Triticum* genome contained the largest number of *RRB* genes among all examined Poaceae genomes, exhibiting a minimum of 12 duplicate events and 2 gene loss events. Both the *Setaria* and maize genomes possessed nine *RRB* genes, having undergone two duplication events each. A heatmap analysis of pairwise protein sequence comparisons among Poaceae *RRB* genes revealed seven clusters, with identities ranging from 26.86% to 99.35% (Figure 1B). The lowest similarity identified was 26.86% between *ZmRRB9* and *SbRRB11*, and the highest similarity was 98.56% between two gene pairs, *TaRRB7*/*TaRRB11* and *TaRRB8*/*TaRRB12*.

### 2.3. Conserved Sequence and Structure Analysis of RRB Genes in Poaceae

We analyzed the conserved motifs and domains of the *RRB* genes to understand their potential functions in Poaceae. Our analysis identified 20 distinct conserved motifs across the 60 RRB proteins (Figure 2A,B). *RRB* members within the same OGCs exhibited similar motif types, arrangements, and numbers. All RRB proteins contained motif 1, which corresponds to the REC_typeB_ARR-like domain. Notably, six types of motifs were found across all seven OGCs, including motifs 1, 3, 5, and 16, corresponding to the REC_typeB_ARR-like domain, and motifs 2 and 4, corresponding to the myb_SHAQKYE domain. In contrast, motifs 8 and 20 appeared exclusively in OGC5, while motifs 11, 12, and 13 were unique to OGC6. Batch CD searches identified four conserved domains in 60 RRB proteins, including the REC_typeB_ARR-like domain (cd17584), the myb_SHAQKYE domain (TIGR01557), the PLN03162 superfamily domain (cl26028), and the PTZ00395 superfamily domain (cl33180) (Figure 2C).

To further determine the structural characteristics and evolutionary patterns of *RRB* genes across various Poaceae species, their cDNA sequences were aligned to their respective genome sequences (Figure 2D). The findings revealed that the numbers of exons in the *RRB* gene family ranged from 4 (*ZmRRB3*) to 8 (*SiRRB6*), with the majority (31 out of 60, 51.7%) containing 6 exons, followed by 24 out of 60 (40.0%) containing 5 exons. This suggests that the last common ancestor (LCA) of the *RRB* gene family in Poaceae likely possessed five or six exons. Nevertheless, most *RRB* genes within the same OGC displayed similar exon–intron organization patterns. Specifically, the *RRB* genes in OGC2 contained six exons, while those in OGC6 and OGC7 had five exons. Additionally, six *RRB* genes in OGC1 (*SbRRB6/7/9/10*, *ZmRRB8*, and *SiRRB4*), seven in OGC3 (*SbRRB5*, *SiRRB2*, *PhRRB2*, *OsRRB3*, and *TaRRB8/10/12*), nine in OGC 4 (*ZmRRB7*, *SbRRB8*, *SiRRB3*, *PhRRB3*, *OsRRB5*, and *TaRRB14/15/17/18*), and six in OGC5 (*SbRRB3*, *SiRRB7*, *OsRRB1*, and *TaRRB1/2/3*) shared similar exon–intron structures. These findings suggest that *RRB* sequences in Poaceae have been evolutionarily conserved, implying that the functions of this gene family are also highly conserved.

### 2.4. Syntenic Relationships and Selective Constraints of the RRB Genes Among the Poaceae Species

To investigate how *RRB* genes expanded across the six species in Poaceae, the chromosomal distribution and duplication events of *RRB* genes were also analyzed (Figure 3 and Figure 4). The results showed that 6 rice *OsRRB* genes are distributed across 4 out of 12 chromosomes. Specifically, one *OsRRB* gene is present on both chromosomes 1 and 3, while two genes are located on each of chromosomes 2 and 6. Synteny analysis identified a total of two segmental duplication gene pairs (*OsRRB2/OsRRB6* and *OsRRB3/OsRRB5*) among the six *OsRRB* genes (Figure 3A). In addition, seven *Panicum PhRRB* genes are located on four out of nine chromosomes, with two, two, one, and two genes on chromosomes 1, 4, 5, and 9, respectively. Among these seven *PhRRB* genes, one segmental duplication gene pair (*PhRRB2/PhRRB3*) was identified (Figure 3B). Furthermore, 11 *Sorghum SbRRB* genes are distributed across 6 out of 10 chromosomes, with chromosome 10 containing the highest number (4 genes). Additionally, chromosomes 1, 3, 4, 5, and 8 contain two, one, two, one, and one *SbRRB* genes, respectively. Two segmental duplication gene pairs (*SbRRB4/SbRRB11* and *SbRRB5/SbRRB8*) and one tandem duplication gene pair (*SbRRB9/SbRRB10*) were identified among the eleven *SbRRB* genes (Figure 3C). Similar findings were observed in the nine *Setaria SiRRB* genes, which are distributed across four out of nine chromosomes. Among these, chromosome 4 contains the highest number, with four genes, while chromosomes 1, 5, and 9 contain two, one, and two genes, respectively. One segmental duplication gene pair (*SiRRB2/SiRRB3)* and one tandem duplication gene pair (*SiRRB4/SiRRB5*) were identified among the nine *SiRRB* genes (Figure 3D). Moreover, 9 maize *ZmRRB* genes are located on 5 out of 10 chromosomes, with 2, 1, 2, 1, and 3 genes on chromosomes 1, 3, 5, 8, and 9, respectively. Three segmental duplication gene pairs (*ZmRRB1/ZmRRB9*, *ZmRRB3/ZmRRB6*, and *ZmRRB5/ZmRRB7*) were identified among the nine *ZmRRB* genes (Figure 3E). Notably, the 18 wheat *TaRRB* genes are distributed across 12 out of 21 chromosomes. Among them, chromosome 7A contains the highest number, with 3 genes; chromosomes 6A, 6B, 6D, and 7D each contain 2 genes, while chromosomes 3A, 3B, 3D, 4A, 4B, 4D, and 7B each contain only 1 gene. A total of 24 segmental duplication gene pairs (*TaRRB1*/*TaRRB2*, *TaRRB1*/*TaRRB3*, *TaRRB2*/*TaRRB3*, *TaRRB4*/*TaRRB5*, *TaRRB4*/*TaRRB6*, *TaRRB5*/*TaRRB6*, *TaRRB7*/*TaRRB9*, *TaRRB7*/*TaRRB11*, *TaRRB8*/*TaRRB10*, *TaRRB8*/*TaRRB12*, *TaRRB8*/*TaRRB15*, *TaRRB8*/*TaRRB16*, *TaRRB8*/*TaRRB17*, *TaRRB9*/*TaRRB11*, *TaRRB10*/*TaRRB12*, *TaRRB10*/*TaRRB15*, *TaRRB10*/*TaRRB16*, *TaRRB10*/*TaRRB17*, *TaRRB12*/*TaRRB15*, *TaRRB12*/*TaRRB16*, *TaRRB12*/*TaRRB17*, *TaRRB15*/*TaRRB16*, *TaRRB15*/*TaRRB17*, and *TaRRB16*/*TaRRB17*) and 2 tandem duplication events (*TaRRB13*/*TaRRB14*/*TaRRB15*, and *TaRRB17*/*TaRRB18*) were identified among the 18 wheat *TaRRB* genes. These distribution and duplication patterns suggest that the *RRB* genes in the six species of Poaceae have evolved through both chromosomal rearrangements and tandem duplication events. These processes might have driven the functional diversification and expansion of this gene family, enhancing the adaptive capabilities of Poaceae.

We further investigated the syntenic relationships between rice *OsRRB* genes and their homologs in five other species, including *Panicum*, *Sorghum*, *Setaria*, maize (*Z. mays*), and wheat (*T. aestivum*). Our analysis revealed that the *OsRRB* genes in rice share 8, 10, 8, 10, and 19 homologous pairs with genes from each of these 5 other species, including *Panicum*, *Sorghum*, *Setaria*, maize, and wheat (Figure 4). Notably, two specific *OsRRB* genes, *OsRRB3*, located on chromosome 2, and *OsRRB5*, located on chromosome 6, showed the highest number of homologous gene pairs with other species, suggestive of their important and conserved roles in the biology of these species in Poaceae. Further, an analysis of non-synonymous (*Ka)* and synonymous (*Ks*) nucleotide substitution rates for these gene pairs between rice and the other five species was calculated. Our analysis revealed that the *Ka/Ks* ratio value was lower than 1 for each gene pair, except for the *OsRRB1/PhRRB5* pair (Table 2). Additionally, the divergence times of these duplication gene pairs ranged from 14.8 to 89.83 MYA (Table 2). These findings display evidence that the evolution of the *RRB* gene family within Poaceae was primarily driven by purifying selection.

To further evaluate the selectivity constraints for OGCs evolution in each *RRB* family, M0 (one-ratio) and M3 (discrete) models were tested for heterogeneity in the *d_N_/d_S_* (*ω*) ratio at codon sites across clusters (Table 3). The mean *ω* ratio under the M0 model was 0.2440 ± 0.0918 (mean ± SD), statistically smaller than 1 but larger than 0 (one-sample *t*-test, *p* < 0.01). This indicated that purifying selection was the dominant evolutionary restraint on the *RRB* family in Poaceae. However, we noticed that the likelihood ratio test (LRT) comparing the discrete model M3 (with three classes of sites) with the null model M0 (which allowed one *ω* ratio across all sites) was statistically notable in all seven OGCs, indicating that *RRB* OGCs differ in their overall level of selected constraints on the codon sites. For further assessment of whether positive selections influenced *RRB* OGCs evolution in Poaceae, two LRT comparisons of M8 (beta and *ω* > 1) vs. M7 (which assumes a beta distribution for *ω* (0 < *ω* < 1)) and M8a (beta and *ω* = 1) vs. M8 models were performed. However, the results of these two LRTs were not significant, and no amino acid site was identified under positive selection, suggesting that none of the seven OGCs underwent positive selection during the evolution of Poaceae.

### 2.5. Identification of Cis-Acting Elements in OsRRB Promoters

*Cis*-acting elements found within promoters are crucial for mediating responses to diverse environmental factors and regulating the expression of specific genes [29]. To figure out these potential roles of Poaceae *RRB* genes for stress response and plant growth and development, we identified the *cis*-acting elements in the promoters of six rice *OsRRB* genes. The distribution and function of these elements are shown in Figure 5A. A total of 30 *cis*-acting elements were discovered across the *OsRRB* promoter regions, which were primarily categorized into four groups: phytohormone response, light response, stress response, growth and development (Figure 5B). All *OsRRBs* contain an anaerobic induction element (ARE) and two MeJA responsive elements (motif-CGTCA- and motif-TGACG). In addition, light-responsive components (G-box and motif-TCT), a gibberellin-responsive element (P-box), a drought-induced element (MBS), a defense- and stress-responsive element (TC-rich repeats), and a meristem expression element (CAT-box) were enriched into more than half of the *OsRRB* genes.

Several other phytohormone-responsive *cis*-elements were identified in certain *OsRRB* members, including auxin-responsive elements (element-TGA and AuxRR-core), a salicylic acid (SA) responsiveness element (element-TCA), an abscisic acid (ABA) responsiveness element (ABRE), and a GA-responsive element (GARE-motif). Additionally, putative stress-responsive *cis*-elements were also found, e.g., the wound-responsive element (WUN-motif) and LTR element. Elements associated with plant growth and development, e.g., the meristem-specific activation element (NON-box) and seed-specific regulation element (RY-element), were detected within several *OsRRB* genes’ promoter regions. These results demonstrated that the *RRB* genes from rice have the potential to play significant roles in phytohormone responses and stress adaptation.

### 2.6. Expression Patterns of the RRB Genes in Rice

Investigating the expression patterns of genes in a large family could offer valuable clues to understanding functional differentiation. We first explored *OsRRB* gene expression across 10 various tissues of rice (Figure 6). Among them, the expression levels of three genes (*OsRRB2*, *OsRRB3*, and *OsRRB5*) were highest in the meristem (Figure 6A). In addition, the expression of *OsRRB1* was highest in the roots, while *OsRRB4* exhibited the highest expression in shoots (Figure 6A). In contrast, *OsRRB6* showed low expression levels across all the tissues examined (Figure 6A). Notably, *OsRRB2* showed relatively higher expression in roots, stems, panicle, meristem, and embryo than the other five *OsRRB* genes, while *OsRRB4* exhibited higher expressions in leaves, seedlings, shoots, seeds, and endosperm than the other genes (Figure 6B).

We also explored the expression patterns of rice *RRBs* in response to diverse phytohormones, including 6-Benzylaminopurine (6-BA, a first-generation synthetic cytokinin), MeJA, and ABA (Figure 7A–C). Under 6-BA treatments, the expression levels of *OsRRB3* and *OsRRB5* were upregulated, while *OsRRB1* expression was down-regulated in comparison with controls. In addition, the expression levels of *OsRRB2* and *OsRRB4* initially decreased before increasing after 6-BA induction, peaking at 12 h. In contrast, the expression levels of *OsRRB6* increased initially and then decreased. Under MeJA treatment, the expression levels of *OsRRB2*, *OsRRB3*, and *OsRRB5* increased initially, then decreased, and subsequently increased again. In contrast, *OsRRB4* expression displayed opposite changes in comparison with the control. Moreover, the expression levels of *OsRRB1* increased before decreasing, while *OsRRB6* expression decreased throughout the treatment. Upon ABA treatment, expression levels of *OsRRB2* to *OsRRB6* increased initially and then decreased. Additionally, *OsRRB1* expression exhibited a more complex pattern, increasing initially, then decreasing, followed by increasing, and subsequently decreasing again.

The expression patterns of the rice *RRB* genes were examined under a variety of abiotic stresses, specifically cold and drought treatments (Figure 7D,E). When exposed to cold stress, the expression levels of *OsRRB1*, *OsRRB2*, *OsRRB5*, and *OsRRB6* increased initially and then decreased. In contrast, both *OsRRB3* and *OsRRB4* expressions decreased initially, then increased, and subsequently decreased again. When exposed to dry stress, expression levels of *OsRRB1* and *OsRRB5* increased initially, then decreased, and subsequently increased again. Moreover, *OsRRB3* expression initially increased before decreasing, while *OsRRB2* exhibited opposite changes. *OsRRB4* and *OsRRB6* expression displayed more complex patterns, initially decreasing, then increasing, followed by another decrease, and finally increasing again.

### 2.7. Natural Variation in OsRRB5 Might Confer Drought Tolerance to Rice

To explore the natural variation in *OsRRB5* in germplasm resources and identity excellent alleles, haplotype analysis was performed using the genotypes from 245 *japonica* rice samples in the 3K rice genome database (Appendix A) and drought tolerance-related phenotypes from the published data [30]. A total of 50 variation sites were detected in both the promoter and coding regions of the *OsRRB5* gene, including 4 insertion–deletion mutations (InDels) and 46 single-nucleotide polymorphisms (SNPs) (Figure 8A). Based on these variations, the 245 rice varieties were classified into 5 distinct haplotypes (Figure 8A,B). Notably, Hap3 exhibited a lower Leaf Color Index (LCI) and Leaf Rolling Index (LRI) compared with the other four haplotypes (Hap1, Hap2, Hap4, and Hap5) under drought stress, but it showed a higher Drought Resistance Index (DRI) than the others (Figure 8C). Further analysis revealed that Hap3 is the only haplotype with the nucleotides “C”, “T”, and “C” at positions −1869 bp, −1293, and −507 upstream of the *OsRRB5* promoter as well as the nucleotide “A” at position 31 in the coding region of *OsRRB5* (Figure 8A). In addition, we noticed that the G to A mutation at position 31 in the coding region results in a non-synonymous amino acid substitution from arginine (Arg, R) to lysine (Lys, K). These findings suggest that these sites in the promoter region might contribute to drought tolerance by affecting the *OsRRB5* gene’s expression or the function of the OsRRB5 protein.

## 3. Discussion

Cytokinin (CK) plays a crucial role in modulating various biotic procedures like plant growth and development together with biotic and abiotic stress responses [1,4,20]. Among the elements of the CK signaling pathway, the response regulator *RR* family is divided into four distinct groups: RRA, RRB, RRC, and PRR [9,17,18]. These groups have been identified across various plant species [8,9,11,12,15,16]. RRBs function as transcription factors that regulate CK-responsive genes, including several *RRA* genes [20]. In *Arabidopsis*, the *RRB* genes have been implicated in various abiotic stress responses, such as drought, salt, and osmotic stresses [3,23,24]. However, in the Poaceae family, the identification and characterization of *RRBs’* functions remain not totally explored. In this work, the genomic-wide identification of 60 *RRB* gene families in selected Poaceae species, the regulatory roles of rice *OsRRB* genes in phytohormonal and abiotic stress responses, and the subcellular localization and haplotypic variation in the *OsRRB5* gene were comprehensively analyzed. Our findings lay the groundwork for further elucidating the functional features of *OsRRB* genes in rice and offer new perspectives for their potential application in rice stress-resistance breeding.

Gene duplication is an important mechanism for generating new genetic material during the evolution of organisms [31]. Among the gene-duplication mechanisms, whole-genome duplication, or polyploidy, is considered a key driver in the evolution of most plants, particularly in numerous Poaceae species [32,33,34,35]. For instance, it is widely accepted that the rice genome underwent two ancient rounds of polyploidy [36], while maize originated as a tetraploid [37,38]. In all, 60 *RRB* genes were identified in the 6 representative species genomes from the Poaceae family. These genes were categorized into 7 OGCs, suggesting that at least 7 *RRB* genes were present from these gramineous species in their common ancestor. Notably, the number of genes within each OGC varied among the six species. This variation was likely owing to gene duplication and/or loss events in the evolutionary history of these six Poaceae species’ genomes. The chromosomal location and gene synteny analyses revealed that the *RRB* genes had 33 pairs of segmental duplications and 4 tandem duplications across these 6 Poaceae species. The ratio of non-synonymous to synonymous substitutions (*ω*), also known as *d_N_/d_S_* or *Ka/Ks*, is commonly used in evolutionary studies to measure the selective constraints on amino acid replacements. A *Ka/Ks* or *d_N_/d_S_* value exceeding 1 indicates positive selection, while below 1 indicates a purification selection [39]. Purifying selection typically reduces genetic diversity, thereby preserving the biological function of natural populations [40]. Our analyses revealed that the majority of gene pairs between rice and the five other species have a *Ka/Ks* < 1. Similarly, *d_N_/d_S_* ratio values for each *RRB* family OGCs were less than 1. In addition, the results of two LRTs comparing M8 vs. M7 and M8a vs. M8 models were not statistically significant. No amino acid sites were found to be under positive selection during the evolution of *RRB* OGCs. These findings provide compelling evidence that purifying selection acts as the primary evolutionary force shaping the *RRB* family in Poaceae.

*Cis*-acting regulatory elements in gene promoters are critical in the transcriptional regulation and functional expression of genes [29]. Here, the identified *cis*-acting elements in rice *OsRRB* genes’ promoters can be categorized into four main groups, including those involved in phytohormone responses, stress responses, and light responses, as well as growth and development. Among the identified *cis*-acting elements, 13 were found to be associated with light responses, implying that *OsRRB* genes are significant in the intricate regulatory networks mediating plant responses to light stimuli. The meristem expression element and meristem-specific activation elements were also identified. Notably, three out of six *OsRRB* genes (*OsRRB2*, *OsRRB3*, and *OsRRB5*) exhibit the highest expression levels in meristem tissue compared with the other nine tissues examined in rice. These findings suggest a potential role for these *OsRRB* genes in meristem development. Additionally, *OsRRB4* displayed relatively higher expression levels in roots, leaves, seedlings, and shoots, while *OsRRB1* showed higher expression levels in roots, stems, and shoots compared with the other tested tissues. These expression patterns imply that *OsRRB1* and *OsRRB4* are involved in rice vegetative growth and development. Given that the *OsRRB6* gene exhibits relatively low expression levels across all examined tissues, its specific biological functions require further investigation. We also noticed that multiple *cis*-acting elements are involved in responses to various phytohormones, including MeJA, GA, ABA, IAA, and SA. Our qRT-PCR analysis also revealed that several *OsRRB* gene expressions can be induced by 6-BA (the first synthetic CK), MeJA, and ABA. Nevertheless, the *OsRRB* genes exhibited distinct expression patterns in response to these phytohormones. For instance, the *OsRRB1*, *OsRRB2*, *OsRRB4*, and *OsRRB6* genes are highly sensitive to these phytohormone treatments, responding rapidly to 6-BA, MeJA, and ABA within a short period of time. In contrast, the expression level of *OsRRB3* was significantly induced by 6-BA treatment after 12 h, while the expression levels of *OsRRB5* were significantly induced by 6-BA and ABA after 6 h of treatment. Given that CK, MeJA, and ABA are key regulators of plant development processes and defense responses [20,41,42], *OsRRB* genes may be involved in various rice physiological responses mediated by these phytohormones or their intricate crosstalk.

Abiotic stresses, such as extreme temperatures and excess or insufficient water, significantly impair plant biochemical and physiological processes, thereby threatening global food security. To cope with these challenges, plants have developed an array of adaptive strategies to thrive in diverse environments, including mechanisms to withstand various abiotic stressors [43]. Notably, several *RRB* genes in *Arabidopsis* are known to be implicated in dealing with multiple abiotic stresses, such as high salinity, water scarcity, and osmotic stresses [3,23,24]. Here, we identified several *cis*-acting elements linked to defense and stress, cold, and drought response within these promoter regions of *OsRRB* genes. Additionally, qRT-PCR analysis revealed that transcriptional levels of some *OsRRB* genes were upregulated in response to cold and drought stresses. However, these duplicated genes showed divergent expression patterns in response to diverse abiotic stresses. The *OsRRB3*, *OsRRB4*, *OsRRB5*, and *OsRRB6* genes respond rapidly to drought and cold treatments within a short period. In contrast, the expression level of *OsRRB1* significantly increased after 9 h of cold treatment, while the expression level of *OsRRB2* was significantly induced by drought treatment after 12 h. From these results, *OsRRB* genes may play an essential role in rice adaptation to abiotic stresses, highlighting their potential significance in enhancing crop resilience to environmental challenges. Importantly, haplotype analysis revealed three key variation sites in the promoter region and one non-synonymous mutation from arginine to lysine in the coding region of the *OsRRB5* gene that may confer drought tolerance in rice. However, since both arginine and lysine are positively charged amino acids with similar physicochemical properties, this mutation is unlikely to significantly alter the three-dimensional structure of the protein or impact its functional properties. Therefore, the most plausible explanation is that the three key variation sites in the promoter region may contribute to drought tolerance by influencing the expression of the *OsRRB5* gene. Further investigation of *OsRRB5*, such as its functions and the mechanisms of its regulation, could provide valuable insights into its significance in rice growth and adaptation to abiotic stresses, particularly drought stresses. Identifying the superior allelic variations in *OsRRB5* and applying them to rice drought-resistance breeding would contribute to global food security.

## 4. Materials and Methods

### 4.1. Identification of RRB Gene Family in Six Poaceae Species

The amino acid sequence of the RRB transcription factor (AT1G67710) served as a query for BLASTP searches against Phytozome database V13 [44]. These BLASTP findings were filtered by applying a 1 × 10^−10^ *E*-value cutoff. Subsequently, the presence of the REC_typeB_ARR-like domain (cd17584) was confirmed using the CD-search tool from the Conserved Domain Database (CDD) at NCBI. Sequences from six representative species of Poaceae, *Oryza sativa*, *Zea mays*, *Sorghum bicolor*, *Panicum hallii*, *Setaria italica*, and *Triticum aestivum*, were collected for further analyses.

### 4.2. Phylogenetic Tree and Sequence Identity Analysis

To construct a phylogenetic tree of 60 *RRB* genes from 6 representative species of Poaceae, multiple protein sequence alignment was conducted by the ClustalW program version 2.0 [45]. The phylogenetic tree was then generated by maximum likelihood (ML) approach utilizing IQ-TREE v.2.3.6 [46]. In this analysis, we determined the optimal model for amino acid substitution and rate heterogeneity using ModelFinder [47]. Branch support values were estimated using ultrafast bootstrap methods [48] with 1000 replicates. Among 298 models estimated, the JTT + F + R3 was identified as the best fit based on Bayesian information criterion (BIC) scores. Additionally, MEGA X v.11.0.2 was used to conduct an NJ (neighbor-joining) phylogenetic tree [49]. The distance analysis employed the JTT model with four rate classifications and estimating gamma distribution factors. Bootstrap analysis was conducted with 1000 replicates. Sequence identity values for the RRB proteins were calculated using Clustal Omega program version 1.2.2 in the EMBL-EBI [50].

### 4.3. Conserved Motif, Domain, and Gene Structure Analysis

Sixty RRB amino acid sequences from six Poaceae species were upload to MEME v5.5.7 [51], where conserved motifs were identified. Additionally, conserved domains of these RRB proteins were identified using Batch CD search in NCBI [52]. GSDS 2.0 [53] was employed to analyze the exon–intron structure of *RRB* genes. The results were visualized with the Gene Structure View program in TBtools v.1.6 [54].

### 4.4. Synteny Analysis of RRB Genes and Ka/Ks Analysis

Tandem duplication genes were identified based on their chromosomal locations. They are characterized by the presence of two or more gene copies arranged consecutively or in close proximity on the same chromosome [55]. Chromosome length data were obtained using the Fasta Stats tool, while gene density information was extracted using the Table Row Extract or Filter tool. The synteny relationships of these segmental duplication gene pairs were analyzed using the One Step MCScanX-Super-fast tool, and the syntenic relationships among the *RRB* genes from each species of Poaceae were visualized with the Advanced Circos tool. Additionally, the syntenic relationships between the rice *OsRRB* genes and their homologs in four other species were analyzed using the One Step MCScanX-Super-fast tool and subsequently visualized with the Dual Synteny Plot tool [54]. All these analyses were performed in TBtools [54]. To assess the selective pressure influencing the evolution of these duplication gene pairs, *Ka* (non-synonymous substitution rate), *Ks* (synonymous substitution rate), and *Ka/Ks* ratio were calculated using TBtools [54]. The approximate data on duplication events were then converted into divergence time (T) for each pair with the equation T = *Ks*/(2 × 1.5 × 10^−8^) × 10^−6^ million years ago (MYA) [56].

### 4.5. Detection of Selective Constraints on Each RRB Orthologous Gene Cluster (OGC)

The protein sequence alignment of each *RRB* OGC was converted into the corresponding codon-based nucleotide alignments via the PAL2NAL program version 1.4 [57]. The aligned protein sequences of RRBs from each OGC were further used to generate the NJ-phylogenetic tree in MEGA X v.11.0.2 [49], respectively. Subsequently, these nucleotide alignments and trees were input into the EasyCodeML program version 1.41 [58]. The likelihood ratio test (LRT) comparing M0 with M3 was employed to assess the heterogeneity in *d_N_/d_S_* (*ω*) across codon sites. Additionally, three other LRTs were conducted under paired site models: M1a vs. M2a, M7 vs. M8, and M8 vs. M8a to detect positive selection. For each LRT, there were two differences in log-likelihood values between these two models that had been compared using the chi-squared (*χ*^2^) statistic. The degrees of freedom were determined by variations observed among the number of factors of the models: 4 for M0/M3 comparison, with 2 for M1a/M2a and M7/M8 trials. The Bayes Empirical Bayes (BEB) approach was applied to model M8 to calculate the posterior probability of sites under positive selection.

### 4.6. Analysis of Cis-Acting Elements Targeting the Promoter of Rice OsRRB Genes

The upstream sequences (2000 bp) of rice *OsRRB* genes were uploaded into the PlantCare promoter database [59] to define their *cis*-acting components. The elements related to phytohormone responses, stress responses, light responses, and plant growth and development were filtered and visualized together with TBtools [54]. A heatmap was created to display *cis*-acting element numbers from each *OsRRB* gene utilizing the HeatMap tool in TBtools [54].

### 4.7. Gene Expression Analysis in Different Tissues of Rice

The gene expressions of *OsRRB* genes were analyzed in 10 various tissues, including the roots, leaves, seedlings, stems, shoots, seeds, panicles, meristems, embryos, and endosperm. Expression levels of six *OsRRB* genes are presented as normalized Log2 (FPKM + 1) values across the different tissues. The FPKM values of 6 *OsRRB* genes in 10 tissues of rice were download from https://plantrnadb.com/ (accessed on 1 November 2024).

### 4.8. Plant Materials and Stress Treatments

A rice variety, Zhonghua 11 (ZH11, *Oryza sativa* L. ssp. *japonica*), was utilized within this study to investigate the expression patterns of *OsRRB* gene in various phytohormones and abiotic stress conditions. ZH11 seedlings were cultivated in nutrient solution in a light incubator maintained at 28 °C, following a 14 h light and 10 h dark cycle for 12 days. For phytohormone treatments, seedlings exhibiting similar growth were treated with 100 µM abscisic acid (ABA) (BBI LIFE SCIENCES, Shanghai, China), methyl jasmonate (MeJA) (Solarbio, Beijing, China), and 6-Benzylaminopurine (6-BA) (Sangon Biotech, Shanghai, China). For cold treatment, 12-day-old seedlings were moved to a new chamber set at 4 °C, following a 14-h light and 10-h dark cycle. For drought treatment, seedlings were exposed to 20% polyethylene glycol 6000 (PEG6000) (Macklin, Shanghai, China). Under the various conditions, collections were made at 0, 1, 3, 6, 9, and 12 h after treatment.

### 4.9. RNA Extraction, cDNA Synthesis, Quantitative Real-Time PCR

The total RNA was extracted following the manufacturer’s instructions with an RNA extraction kit (Tiangen Biotech, Beijing, China). First-strand cDNA was synthesized by extracting approximately 1 μg of RNA from each sample with HiScript III First Strand cDNA Synthesis Kit (+gDNA Wiper) (Vazyme Biotech, Nanjing, China). *RRB* gene family from rice was characterized by Quantitative Real-Time PCR (qRT-PCR) analysis with Hieff qPCR SYBR Green Master Mix (No Rox) (Yeasen Biotech, Shanghai, China). The relative expression levels of six rice *OsRRB* genes were evaluated following the 2^−△△CT^ method [60]. The relative expression levels of target genes were normalized using *OsActin* (gene locus: LOC_Os03g50885) as an internal control gene. The qRT-PCR assays were conducted using three biological replicates. The specific elicitors for the rice *OsRRB* genes utilized in qRT-PCR analysis are provided in Appendix A. The quantification cycle (Cq) values in qRT-PCR analysis are presented in Appendix A. Amplification curves, melting curves, and standard curves for *OsRRB* and *OsActin* genes in qRT-PCR analysis are illustrated in Appendix A, respectively.

### 4.10. Haplotype Analysis of OsRRB5 Genes

The genomic sequences of 245 rice varieties were obtained from the Rice Functional Genomics and Breeding Database [61,62]. Haplotype analyses were performed using DNA Sequence Polymorphism (DnaSP) v6 [63]. Drought tolerance-related phenotypes, including Leaf Color Index (LCI), Leaf Rolling Index (LRI), and Drought Resistance Index (DRI) were obtained from previously published data [30].

### 4.11. Data Analyses

The statistical analysis to detect significant differences was carried out with Tukey’s HSD test (*p* < 0.05) employing IBM SPSS software v25.

## 5. Conclusions

In summary, a total of 60 *RRB* genes from 6 Poaceae species were classified into 7 OGCs deriving from the phylogenetic tree and syntenic analyses, indicating that the common ancestor of these Poaceae species possessed at least 7 *RRB* genes, with the observed variations in gene numbers across these species resulting from gene duplication and/or loss events. Further analyses indicated that the evolution of the *RRB* family gene in Poaceae was primarily driven by purifying selection. The transcriptional levels of these genes can be induced by various phytohormones, including CK, MeJA, and ABA, as well as abiotic factors such as cold and drought. Importantly, haplotype analysis identified four key variation sites in the *OsRRB5* genomic regions that may confer drought tolerance in rice. Our findings lay the groundwork for further elucidating the biological function of rice *OsRRB* genes and provide a promising new gene target for rice stress-resistance breeding.

## Figures and Tables

**Figure 1 ijms-26-01954-f001:**
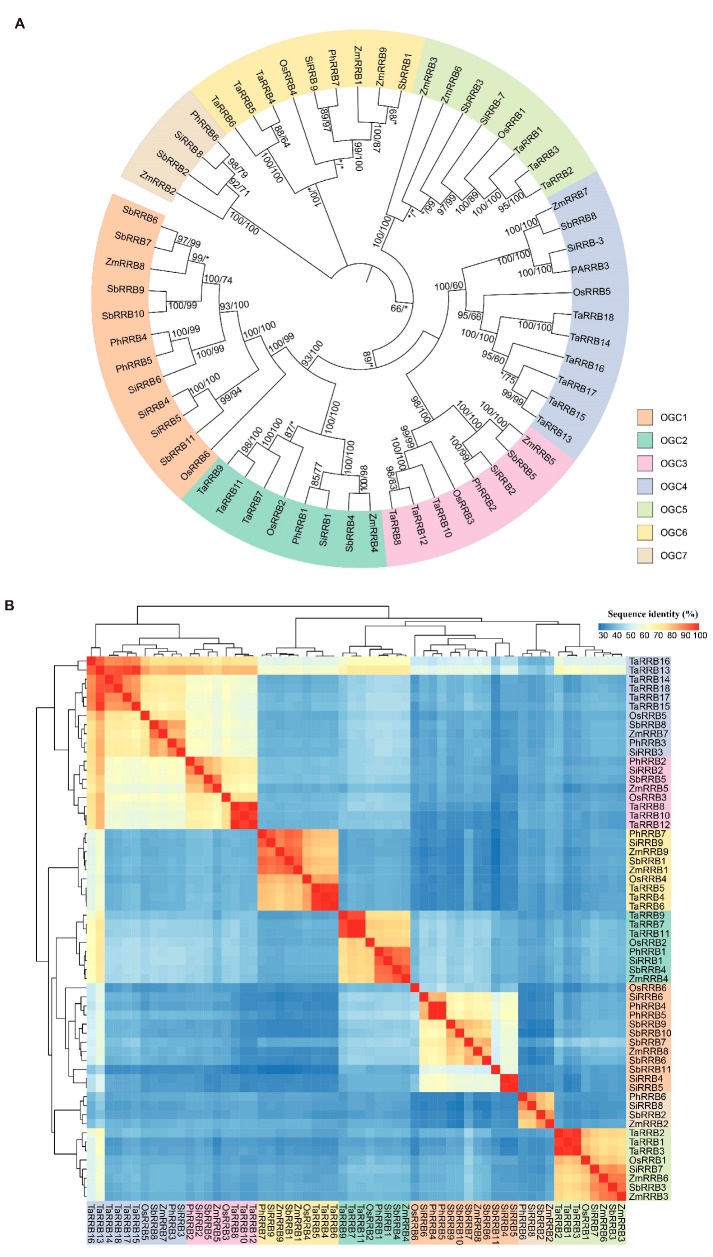
Phylogenetic tree and sequence identity analysis of *RRB* genes in Poaceae. (**A**) The phylogenetic tree was constructed using Jones–Taylor–Thornton (JTT) substitution models in both maximum likelihood (ML) and distance analyses. Numbers on the branches represent bootstrap percentage values from both methods, while asterisks indicate values less than 50%. The colored clades highlight diverse orthologous gene clusters (OGCs). (**B**) A heatmap diagram displays the sequence identity values among Poaceae RRB proteins.

**Figure 2 ijms-26-01954-f002:**
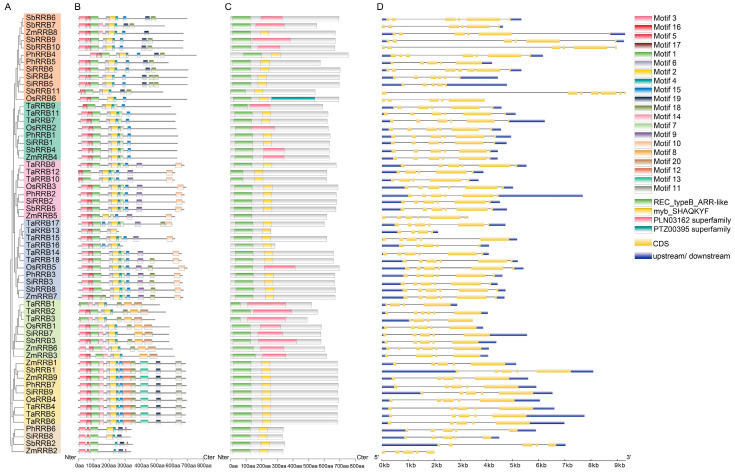
The conserved motif, domain, and gene structure analysis of *RRB* genes in Poaceae. (**A**) The phylogenetic tree of Poaceae *RRB* genes using the ML method. (**B**) The distribution of conserved motifs in the RRB proteins. The colored boxes represent motifs 1–20. Among them, motifs 1, 3, 5, and 16 correspond to the REC_typeB_ARR-like domain, while motifs 2 and 4 correspond to the myb_SHAQKYE domain. Nter, N-terminal; Cter, C-terminal. (**C**) The conserved domains of RRB proteins. The colored boxes represent distinct domains. Nter, N-terminal; Cter, C-terminal. (**D**) The gene structure of *RRB* genes. Orange boxes indicate the exons, black lines represent introns, and blue boxes indicate the 5′-UTR and 3′-UTR.

**Figure 3 ijms-26-01954-f003:**
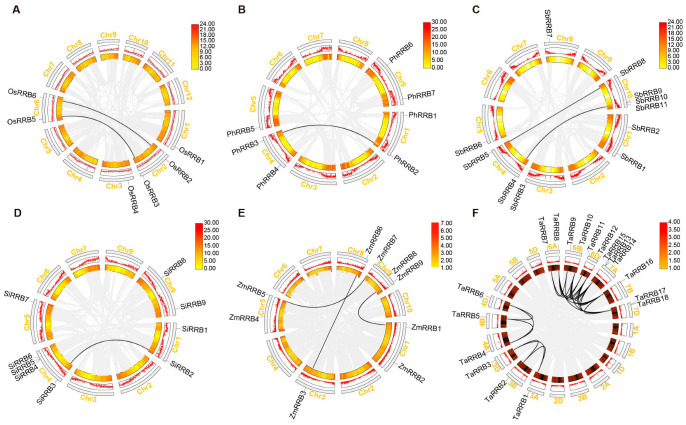
The chromosomal locations and duplication events of *RRBs* in six species of Poaceae. The positions of *RRB* genes from six species, *Oryza sativa* (**A**), *Panicum hallii* (**B**), *Sorghum bicolor* (**C**), *Setaria italica* (**D**), *Zea may*s (**E**), and *Triticum aestivum* (**F**) are marked. The segmental duplication gene pairs are connected by black lines. Heatmaps indicate gene density, while the gray curved boxes represent the different chromosomes.

**Figure 4 ijms-26-01954-f004:**
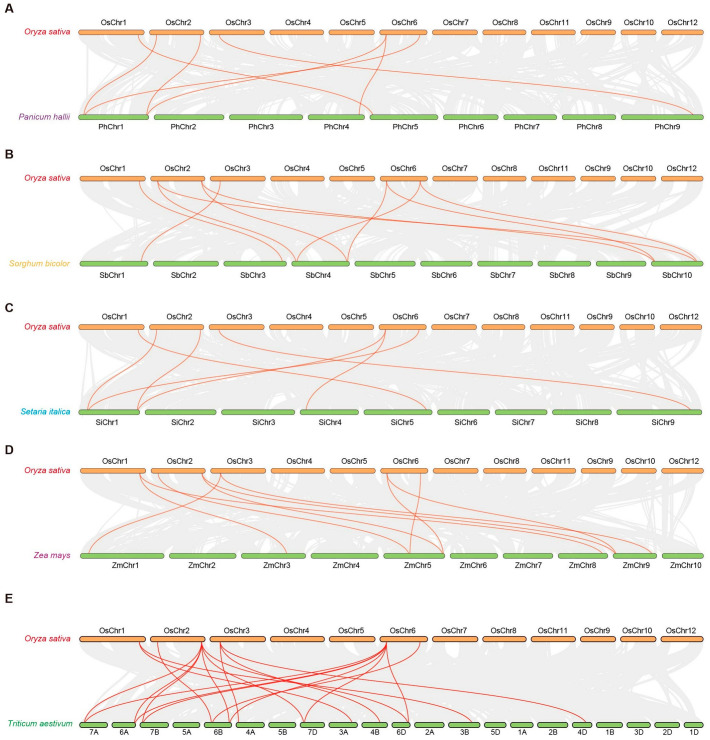
Syntenic analysis of *RRB* genes between rice (*Oryza sativa*) and the other five species of Poaceae. (**A**) The synteny analysis of *RRB* genes between rice and *P. hallii*. (**B**) The synteny analysis of *RRB* genes between rice and *S. bicolor*. (**C**) The synteny analysis of *RRB* genes between rice and *S. italica*. (**D**) The synteny analysis of *RRB* genes between rice and *Z. mays.* (**E**) The synteny analysis of *RRB* genes between rice and *T. aestivum.* Gray lines in the background represent the collinear blocks within rice and the other five species, while red lines indicate homologous gene pairs.

**Figure 5 ijms-26-01954-f005:**
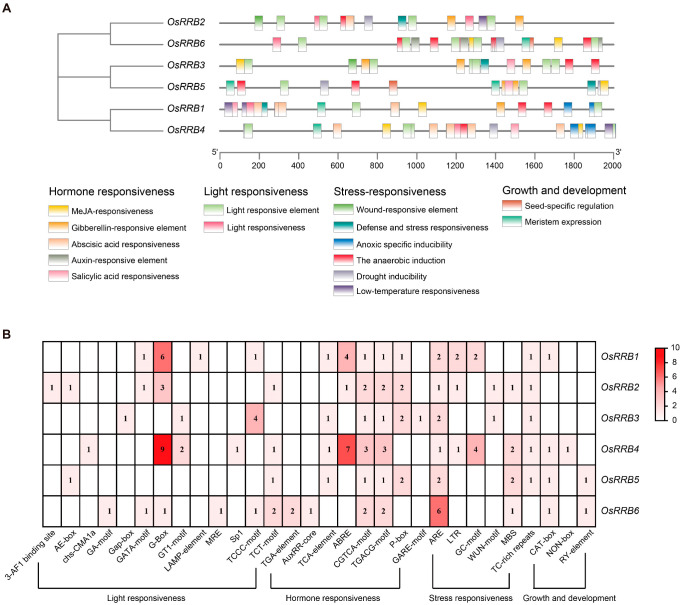
*Cis*-acting element analysis of the promoter regions of *OsRRB* genes. (**A**) Distribution of *cis*-acting elements classified by functions in *OsRRBs*. The promoter sequences with 2000 bps are analyzed. Different-color boxes represent various functional categories of *cis*-acting elements. (**B**) A heatmap diagram illustrates the number of each *cis*-acting element in *OsRRBs*.

**Figure 6 ijms-26-01954-f006:**
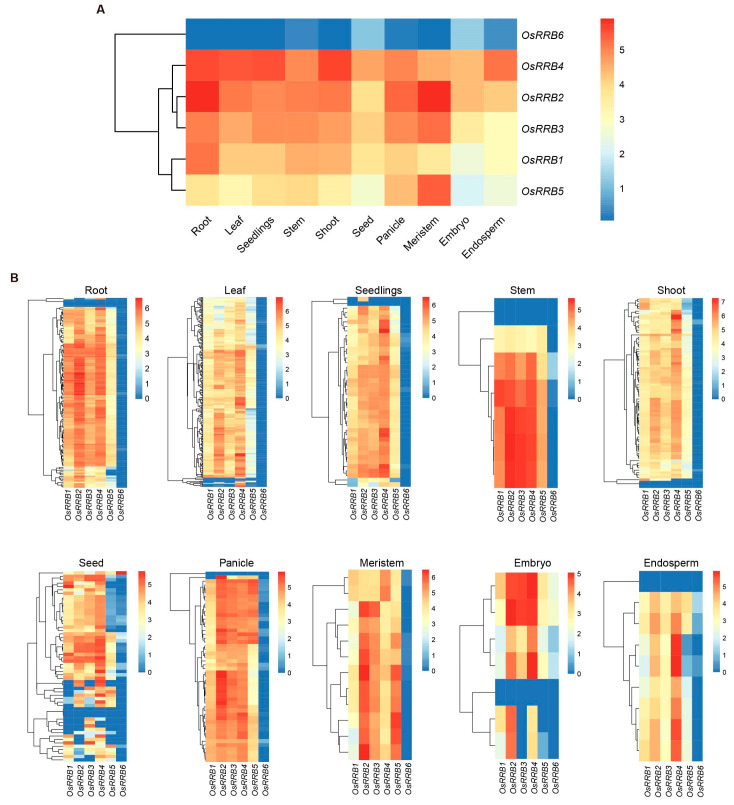
Expression patterns of *OsRRB* genes in various tissues of rice. (**A**) Gene expression patterns of 6 *OsRRB* genes in 10 tissues of rice. Expression levels are presented as normalized Log2 (FPKM mean + 1) values across different tissues. The mean FPKM values were calculated from multiple tissue samples: roots (*n* = 125), leaves (*n* = 242), seedlings (*n* =42), stems (*n* = 7), shoots (*n* = 80), seeds (*n* = 56), panicles (*n* = 50), meristems (*n* = 12), embryos (*n* = 7), and endosperm (*n* = 9). (**B**) Expression levels of 6 *OsRRB* genes are presented as normalized Log2 (FPKM + 1) values across 10 different tissues. The analysis incorporated data from the same tissues and n numbers as in (**A**). The FPKM values of six *OsRRB* genes were download from https://plantrnadb.com/ (accessed on 1 November 2024).

**Figure 7 ijms-26-01954-f007:**
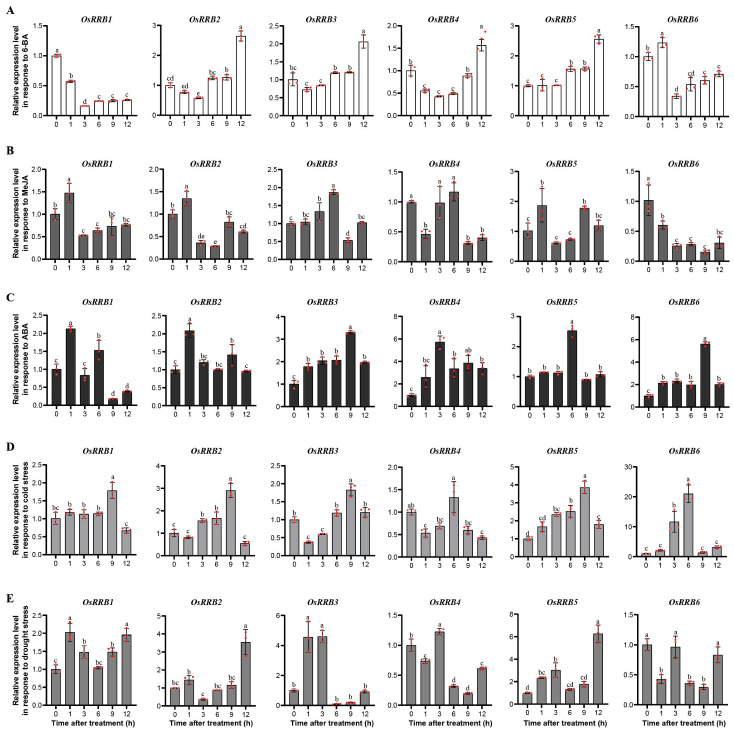
Expression patterns of rice *OsRRB* genes in response to diverse phytohormones and abiotic stresses. (**A**–**C**) Gene expression patterns of six *OsRRB* genes in response to 6-Benzylaminopurine (6-BA) (**A**), methyl jasmonate (MeJA) (**B**), and abscisic acid (ABA) (**C**). (**D**,**E**) Gene expression patterns of six *OsRRB* genes in response to cold (**D**) and drought (**E**) stresses. *OsActin* was used as a control. Data show means ± SD (*n* = 3). Different letters indicate statistically significant differences at *p* < 0.05, as determined by a one-way ANOVA test.

**Figure 8 ijms-26-01954-f008:**
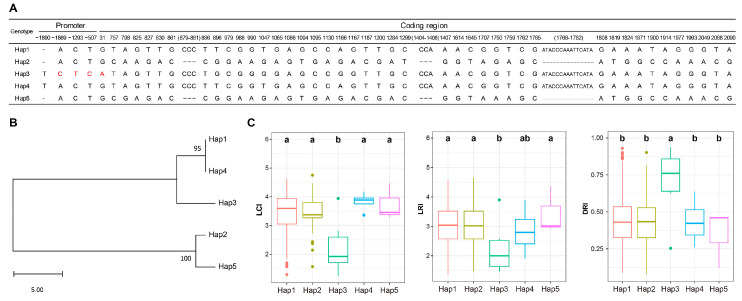
Haplotype analysis of *OsRRB5* gene in japonica rice population. (**A**) Haplotypes of *OsRRB5* in natural rice population. Nucleotide variations in the promoter and coding regions of *OsRRB5* are shown. Red fonts indicate the key variations in the promoter and coding region of *OsRRB5*. (**B**) Phylogenetic tree of the five haplotypes of *OsRRB5*. (**C**) Comparison of the Leaf Color Index (LCI), Leaf Rolling Index (LRI), and Drought Resistance Index (DRI) among rice varieties with the five haplotypes: *n* = 189 in Hap1, *n* = 43 in Hap2, *n*= 6 in Hap3, *n* = 4 in Hap4, and *n* = 3 in Hap5. Different letters indicate statistically significant differences at *p* < 0.05, as determined by a one-way ANOVA test.

**Table 1 ijms-26-01954-t001:** Physicochemical properties and subcellular localizations of RRB proteins in six species of Poaceae.

Gene ID	Gene Name	Protein Length (aa)	Molecular Weight (Da)	Theoretical Isoelectric Point	Instability Index	Aliphatic Index	Grand Average of Hydropathicity	Predicted Subcellular Localization
LOC_Os01g67770	*OsRRB1/OsRR26*	582	64,813.48	5.04	52.76	83.21	−0.426	nucl
LOC_Os02g08500	*OsRRB2/OsRR24*	626	68,399.04	6.06	35.29	75.85	−0.602	nucl
LOC_Os02g55320	*OsRRB3/OsRR23*	688	74,383.99	6.45	49.99	81.35	−0.323	nucl
LOC_Os03g12350	*OsRRB4/OsRR21*	691	73,807.04	6.06	46.47	75.77	−0.399	nucl
LOC_Os06g08440	*OsRRB5/OsRR22*	696	75,949.63	6.01	48.51	82.84	−0.406	nucl
LOC_Os06g43910	*OsRRB6/OsRR25*	694	76,623.66	5.65	46.09	62.13	−0.544	nucl
Pahal.A00557	*PhRRB1*	637	69,308.4	6.03	38.74	77.43	−0.548	nucl
Pahal.A03697	*PhRRB2*	677	73,794.06	5.93	50.02	83.81	−0.349	nucl
Pahal.F02357	*PhRRB3*	667	72,992.9	6.01	49.84	79.24	−0.358	nucl
Pahal.D00864	*PhRRB4*	753	81,469.49	6.02	43.27	75.27	−0.427	chlo
Pahal.E00452	*PhRRB5*	575	64,419.32	5.2	44.71	81.17	−0.427	nucl
Pahal.I03137	*PhRRB6*	338	38,692.6	5.72	45.69	81.63	−0.517	nucl
Pahal.B00059	*PhRRB7*	689	73,949.03	6.23	48.73	77.85	−0.437	nucl
Sobic.001G451000	*SbRRB1*	686	73,679.76	6.26	49.94	77.76	−0.423	nucl
Sobic.001G227900	*SbRRB2*	347	39,281.32	5.57	45.99	84.55	−0.404	nucl
Sobic.003G393300	*SbRRB3*	579	64,700.22	5.04	45.88	81.3	−0.424	nucl
Sobic.004G066600	*SbRRB4*	631	68,631.82	6.12	39.97	77.4	−0.533	nucl
Sobic.004G330900	*SbRRB5*	675	73,136.19	5.96	46.1	80.99	−0.341	nucl
Sobic.005G050700	*SbRRB6*	694	75,483.58	5.78	39.05	77.8	−0.45	nucl
Sobic.008G071200	*SbRRB7*	551	60,852.22	5.66	36.22	80.69	−0.536	nucl
Sobic.010G064700	*SbRRB8*	672	72,886.35	6.01	46.69	80.45	−0.331	nucl
Sobic.010G191900	*SbRRB9*	671	72,439.25	6.71	45.97	73.53	−0.395	nucl
Sobic.010G192200	*SbRRB10*	668	72,765.06	6.58	42.33	77.65	−0.354	nucl
Sobic.010G208100	*SbRRB11*	531	58,488.74	6.08	44.59	71.02	−0.522	vacu
Seita.1G061200	*SiRRB1*	634	69,271.5	6.16	44.78	76.28	−0.578	nucl
Seita.1G351300	*SiRRB2*	679	73,556.54	6.24	46.58	79.25	−0.38	nucl
Seita.4G050700	*SiRRB3*	666	72,711.52	6.13	45.16	79.07	−0.321	nucl
Seita.4G198500	*SiRRB4*	696	74,927.51	6.06	49.66	71.29	−0.41	nucl
Seita.4G198600	*SiRRB5*	696	74,956.4	6.02	50.71	71.29	−0.415	nucl
Seita.4G215900	*SiRRB6*	701	75,701.3	5.46	38.48	71.37	−0.485	nucl
Seita.5G419200	*SiRRB7*	578	64,841.78	5.12	48.87	82.92	−0.433	nucl
Seita.9G231000	*SiRRB8*	332	38,099.01	5.76	40.24	81.33	−0.502	nucl
Seita.9G485100	*SiRRB9*	687	73,746.87	6.38	49.82	76.26	−0.432	nucl
Zm00001d028265	*ZmRRB1*	700	75,325.64	6.16	49.28	77.86	−0.38	nucl
Zm00001d032784	*ZmRRB2*	337	38,292.12	5.32	45.25	82.43	−0.465	nucl
Zm00001d042463	*ZmRRB3*	615	69,249.43	5.13	45.98	80.93	−0.424	nucl
Zm00001d015521	*ZmRRB4*	631	68,637.9	6.25	37.13	80.02	−0.493	nucl
Zm00001d018380	*ZmRRB5*	616	66,308.81	6.13	47.92	83.57	−0.25	nucl
Zm00001d012128	*ZmRRB6*	603	66,962.82	5.04	45.1	81.92	−0.4	nucl
Zm00001d045112	*ZmRRB7*	669	72,502.03	5.91	45.11	81.08	−0.311	nucl
Zm00001d046755	*ZmRRB8*	671	73,467.82	7.18	44.91	79.57	−0.406	nucl
Zm00001d048046	*ZmRRB9*	686	74,002.02	6.12	45.98	77.03	−0.435	pero
TraesCS3A02G391600	*TaRRB1*	519	57,469.68	5.3	57.46	71.75	−0.501	nucl
TraesCS3B02G423600	*TaRRB2*	558	61,882.04	5.46	52.54	77.03	−0.417	nucl
TraesCS3D02G384500	*TaRRB3*	491	54,106.03	5.57	56.22	73.65	−0.47	nucl
TraesCS4A02G063100	*TaRRB4*	684	73,659.79	6.2	47.16	75.28	−0.436	nucl
TraesCS4B02G240100	*TaRRB5*	684	73,606.65	6.16	47.1	75.41	−0.434	nucl
TraesCS4D02G239900	*TaRRB6*	684	73,603.69	6.16	47.52	75.15	−0.443	nucl
TraesCS6A02G146200	*TaRRB7*	619	67,064.32	6.11	36.97	72.6	−0.606	nucl
TraesCS6A02G359200	*TaRRB8*	676	72,702.61	6.17	50.36	77.35	−0.322	chlo
TraesCS6B02G174400	*TaRRB9*	591	64,329.17	6.02	39.71	71.12	−0.609	nucl
TraesCS6B02G392000	*TaRRB10*	616	66,160.07	5.91	50.9	75.55	−0.348	nucl
TraesCS6D02G135500	*TaRRB11*	624	67,472.72	6.11	39.16	72.8	−0.61	nucl
TraesCS6D02G342200	*TaRRB12*	617	66,248.1	5.87	52.72	75.27	−0.359	nucl
TraesCS7A02G146500	*TaRRB13*	258	29,667.3	6.76	42.28	92.52	−0.443	cyto
TraesCS7A02G146700	*TaRRB14*	659	72,307.75	5.84	46.41	82.29	−0.341	nucl
TraesCS7A02G146400	*TaRRB15*	618	68,018.19	6.52	42.19	83.79	−0.332	nucl
TraesCS7B02G049000	*TaRRB16*	284	32,613.36	6.73	36.72	87.11	−0.582	nucl
TraesCS7D02G148000	*TaRRB17*	601	66,309.09	5.81	42.68	82.95	−0.363	nucl
TraesCS7D02G148200	*TaRRB18*	659	72,017.68	5.92	49.59	84.07	−0.311	nucl

Note: nucl, nucleus; vacu, vacuole; pero, peroxisome; chlo, chloroplast; cyto, cytoplasm.

**Table 2 ijms-26-01954-t002:** *Ka/Ks* values and divergence time of all duplication gene pairs between rice *OsRRB* genes and the other four species of Poaceae.

Seq_1	Seq_2	*Ka*	*Ks*	*Ka/Ks*	Data (MYA)
*OsRRB1*	*PhRRB5*	3.4173	2.6949	1.2681	89.83
*OsRRB2*	*PhRRB1*	0.1274	0.5719	0.2227	19.06
*OsRRB3*	*PhRRB2*	0.1539	0.6039	0.2548	20.13
*OsRRB3*	*PhRRB3*	0.231	0.8917	0.259	29.72
*OsRRB4*	*PhRRB7*	0.0883	0.553	0.1598	18.43
*OsRRB5*	*PhRRB2*	0.2535	1.02	0.2485	34.00
*OsRRB5*	*PhRRB3*	0.1416	0.6974	0.2031	23.25
*OsRRB6*	*PhRRB1*	0.3996	1.2841	0.3112	42.80
*OsRRB1*	*SbRRB3*	0.1119	0.5629	0.1989	18.76
*OsRRB2*	*SbRRB4*	0.1275	0.6276	0.2031	20.92
*OsRRB2*	*SbRRB11*	0.5467	1.4756	0.3705	49.19
*OsRRB3*	*SbRRB5*	0.1599	0.6731	0.2375	22.44
*OsRRB3*	*SbRRB8*	0.2332	0.8747	0.2666	29.16
*OsRRB4*	*SbRRB1*	0.0889	0.6128	0.145	20.43
*OsRRB5*	*SbRRB5*	0.2487	1.093	0.2275	36.43
*OsRRB5*	*SbRRB8*	0.1519	0.7098	0.2141	23.66
*OsRRB6*	*SbRRB4*	0.4002	1.117	0.3583	37.23
*OsRRB6*	*SbRRB11*	0.5331	1.2526	0.4256	41.75
*OsRRB1*	*SiRRB7*	0.1047	0.7002	0.1496	23.34
*OsRRB2*	*SiRRB1*	0.1216	0.6481	0.1877	21.60
*OsRRB3*	*SiRRB2*	0.1538	0.5902	0.2605	19.67
*OsRRB3*	*SiRRB3*	0.2397	0.9709	0.2468	32.36
*OsRRB4*	*SiRRB9*	0.0955	0.5807	0.1644	19.36
*OsRRB5*	*SiRRB2*	0.2468	0.9818	0.2514	32.73
*OsRRB5*	*SiRRB3*	0.1477	0.6568	0.225	21.89
*OsRRB6*	*SiRRB1*	0.4103	1.3571	0.3023	45.24
*OsRRB1*	*ZmRRB3*	0.1252	0.5894	0.2124	19.65
*OsRRB1*	*ZmRRB6*	0.1177	0.5639	0.2088	18.80
*OsRRB2*	*ZmRRB4*	0.1255	0.681	0.1843	22.70
*OsRRB3*	*ZmRRB5*	0.1815	0.6716	0.2702	22.39
*OsRRB3*	*ZmRRB7*	0.247	1.1929	0.2071	39.76
*OsRRB4*	*ZmRRB1*	0.0971	0.6635	0.1463	22.12
*OsRRB4*	*ZmRRB9*	0.1032	0.639	0.1615	21.30
*OsRRB5*	*ZmRRB5*	0.2794	1.1734	0.2381	39.11
*OsRRB5*	*ZmRRB7*	0.1647	0.9428	0.1747	31.43
*OsRRB6*	*ZmRRB4*	0.4282	1.2706	0.337	42.35
*OsRRB1*	*TaRRB2*	0.1006	0.6258	0.1607	20.86
*OsRRB4*	*TaRRB4*	0.0777	0.4441	0.1749	14.80
*OsRRB4*	*TaRRB5*	0.082	0.4541	0.1805	15.14
*OsRRB4*	*TaRRB6*	0.0812	0.4712	0.1723	15.71
*OsRRB3*	*TaRRB17*	0.2508	1.249	0.2008	41.63
*OsRRB5*	*TaRRB17*	0.1651	0.8753	0.1887	29.18
*OsRRB3*	*TaRRB8*	0.1297	0.55	0.2359	18.33
*OsRRB5*	*TaRRB8*	0.2573	1.0863	0.2369	36.21
*OsRRB1*	*TaRRB1*	0.11	0.6795	0.1619	22.65
*OsRRB3*	*TaRRB15*	0.2403	1.1425	0.2103	38.08
*OsRRB5*	*TaRRB15*	0.1785	0.8258	0.2161	27.53
*OsRRB3*	*TaRRB16*	0.1448	1.4261	0.1015	47.54
*OsRRB5*	*TaRRB16*	0.1133	1.0021	0.113	33.40
*OsRRB2*	*TaRRB9*	0.123	0.6119	0.2009	20.40
*OsRRB6*	*TaRRB9*	0.4317	1.4272	0.3025	47.57
*OsRRB3*	*TaRRB10*	0.1363	0.6138	0.222	20.46
*OsRRB5*	*TaRRB10*	0.2723	1.2458	0.2185	41.53
*OsRRB3*	*TaRRB12*	0.1309	0.588	0.2226	19.60
*OsRRB5*	*TaRRB12*	0.2671	1.1259	0.2372	37.53

Note: Ka, non-synonymous substitution rate; Ks, synonymous substitution rate; Ka/Ks, the ratio of non-synonymous substitution rate and synonymous substitution rate; MYA, million years ago.

**Table 3 ijms-26-01954-t003:** Selective pressure analysis of each *RRB* OGC in Poaceae.

Group	*N*	*d_N_/d_S_* (*ω*) Under M0	2Δ*InL*, M3 vs. M0	2Δ*InL*, M8 vs. M7	2Δ*InL*, M8 vs. M8a	Parameter Estimates Under M8	Positively Selected Sites
OGC1	12	0.4033	199.6352 **	1.0967	−2.2265	*p*1 = 0.0000, *ω* = 1.0000 *β* (*p* = 0.2460, *q* = 1.0868)	NAN
OGC2	8	0.2076	62.7116 **	−0.0001	0.7052	*p*1 = 0.0000, *ω* = 1.0000 *β* (*p* = 0.3902, *q* = 1.2106)	NAN
OGC3	8	0.2758	111.4799 **	−0.0022	0.2532	*p*1 = 0.0000, *ω* = 2.6422 *β* (*p* = 0.4916, *q* = 1.0399)	NAN
OGC4	11	0.1082	45.3988 **	−0.0006	2.4848	*p*1 = 0.0000, *ω* = 1.0000 *β* (*p* = 0.1741, *q* = 1.0287)	NAN
OGC5	8	0.2206	69.5820 **	0.8325	0.0546	*p*1 = 0.0816, *ω* = 1.0000 *β* (*p* = 0.4221, *q* = 1.7059)	NAN
OGC6	9	0.2023	96.9320 **	−0.0001	1.0639	*p*1 = 0.0000, *ω* = 1.0000 *β* (*p* = 0.2726, *q* = 0.8832)	NAN
OGC7	4	0.2902	29.8178 **	−0.0002	7.8491	*p*1 = 0.0044, *ω* = 998.9998 *β* (*p* = 0.3026, *q* = 0.6254)	NAN

Note: OGC, orthologous gene cluster; *N*, the number of genes in each OGC; *ω = d_N_/d_S_*, the ratio of non-synonymous substitution rate and synonymous substitution rate; M0, one ratio model; M3, the discrete model; M7, the beta distribution model (0 < *ω* < 1); M8, the beta model with *ω* > 1; M8a, the beta model with *ω* = 1; 2Δ*InL*, two differences in log-likelihood values between the null model and alternative model. For each likelihood ratio test (LRT), the chi-squared (χ^2^) test (**, *p* < 0.01) was used for statistical analysis. A beta distribution with the parameters *p* and *q* was used to describe the variable *ω* for sites within the range of 0 < *ω* < 1. *p*0 represents the proportion of sites with *ω* from beta (*p*, *q*), while *p1* is defined as 1 − *p*0. NAN denotes that no positively selected sites were detected.

## Data Availability

All the data that support the findings of this study are available in the paper and its Appendix A published online.

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
