# Peer review of "Comparative Genomic Analysis of the Poaceae Cytokinin Response Regulator RRB Gene Family and Functional Characterization of OsRRB5 in Drought Stress Tolerance in Rice"

_ijms, 2025, doi:10.3390/ijms26051954_

Round 1

Reviewer 1 Report

Comments and Suggestions for Authors

The manuscript presents an extensive analysis of the phylogenetic relationships, evolutionary pressures, and expression patterns of the ARR-B gene family in five grass species (Gramineae), as well as evaluates the expression pattern of some of these genes in rice under different stresses and exposure to plant growth regulators.
The paper is generally well-written and provides relevant information for the field.
On the other hand, there are some points that need to be better presented and corrected, if necessary, as outlined below:

1) The discussion is too general and lacks argumentation related to the results obtained. For example, what is the influence of sample collection times on the observed gene expression pattern? Are there genes that respond earlier to stress or hormonal induction?

Conduct a discussion on gene synteny based on the different chromosomal numbers of the species studied (Oryza sativa n=12, Panicum hallii n=9, Sorghum bicolor n=10, Setaria italica n=9, Zea mays n=10).

The results of gene expression patterns observed across different tissues were not discussed.

2) The number of biological and technical replicates used in the experiment was not presented in the Materials and Methods section. This information needs to be included. See recommendations from the MIQE (The Minimum Information for Publication of Quantitative Real-Time PCR Experiments) guidelines [Bustin, S.A.; Benes, V.; Garson, J.A.; Hellemans, J.; Huggett, J.; Kubista, M.; Mueller, R.; Nolan, T.; Pfaffl, M.W.; Shipley, G.L.; Vandesompele, J.; Wittwer, C.T. The MIQE guidelines: minimum information for publication of quantitative real-time PCR experiments. Clin. Chem. 2009, 55, 611–622.] https://doi.org/10.1373/clinchem.2008.112797

3) Which normalizer genes were used in the qPCR?

4) Include in the supplementary information the Cycle of Quantification (Cq) data obtained from RT-qPCR reactions; amplification curves for both target and reference genes; melting curves; and efficiency curves.

5) Figure 3, line 222. How can there be tandem duplication on different chromosomes? The black lines indicating this (tandem duplication) are not appropriate.

6) Page 9, line 224. Correct the epithet of the species name S. italic (S. italica).

7) Page 13, lines 244-248. This is methodology and should be placed in the Materials and Methods section.

8) Page 14, lines 279-280. This statement must be included in the discussion. In the results section, present the data that were obtained, avoiding any discussion.

9) Page 18, line 430. Correct the epithet of the species name Setaria italic (Setarua italica).

10) Page 19, lines 452-453. Why was 70% sequence identity used to define tandem gene duplication? Reference [28] is not appropriate.

11) Page 20, lines 502-503. How many biological and technical replicates were used?

Reviewer 2 Report

Comments and Suggestions for Authors

The authors performed a global and comprehensive analysis of ARR-B in 5 Poaceae species with a focus on RR-B5 in rice.

As a general comment, the manuscript is correctly written and presents an important collection of data. Nevertheless, the manuscript suffers from a lack of citations since there’s a lot of articles that have been published concerning TCS members in Poaceae species such as rice, maize and wheat. The experiments are only described and poorly interpreted. The discussion should be developed and include more references, as well as the introduction.

Below is a list of major points that constitute a prerequisite prior any publication:

- The term “Gramineae” throughout the text is no longer in use and has to be replaced by the official term “Poaceae”.

- The nomenclature of TCS members has been settled by a consortium of international scientists working on CK signaling a long time ago (Heyl et al., 2013, Plant Physiol), therefore the right naming of OsRR should be used. At least, the correspondence with other names of RR-B should be given.

- There’s not only 2 types of RR but 4: RRA, RRB, RRC and PRR. A lot of publications describe these 4 types and this should be implemented in the introduction.

- There’s a lot of publications on TCS members in rice, maize and wheat:

Rice: Pareek et al., 2006; Ito et al., 2006 and more recently Saha et al., 2024 with the exact same analyses on rice RR-B.

Maize: Asakura et al., 2003

Wheat: Pan et al., 2009 ; Gahlaut et al., 2014

Evolution of TCS members : Pils & Heyl, 2009

Those references, and much more, constitute already a huge amount of data on Poaceae species and they are not mentioned in this article.

- The term “TCS” is appropriate to describe prokaryote system with 2 components. For eukaryote system, the term “MSP” (Multi-Step Phosphorelay) is more accurate since there’s more than 2 components. Accordingly, “TCS” should be replaced by “MSP”. Read Mira-Rodado, 2019 (Plants) for a review on the comparison bacteria/plants.

- The figures 1 and 2 are too small. It’s impossible to read properly.

- All the tables have no legend. The abbreviations used in these tables have to be explained.

- The Fig. 2 presents different motifs found in RR genes but there’s no explanation about these motifs, except their number. This should be added.

- The paragraph 2.5 is very short and completely related to the data presented in paragraph 2.4. These 2 paragraphs should be fused in one.

- Since the notion of positive or negative selection and statistical models to calculate it is not easy to understand and not familiar to all scientists, these points should be much more explained in the text (results section as well as discussion section).

- Why the authors didn’t choose Wheat as a plant model for this study since its genome has been fully sequenced in 2018?

- In paragraph 2.7, the authors analyzed the expression patterns of OsRR-B genes in different tissues. Where this information comes from? Is it from public available data or it corresponds to authors data? There’s no explanation about the origin of this experiment.

- In the haplotype analysis (Fig. 8), a mutation at position 31 has been identified. What is the consequence of this mutation in the coding sequence? Is it a silent mutation or a substitution? If so, what is the changed residue? What could be the impact of this mutated allele on the protein? This should be presented and discussed.

Below is a list of minor points that have to be corrected:

- L267: the assertion should be moderated. Replace “…gene from rice play significant…” by “…genes from rice have the potential to play significant…”.

- L277: avoid useless repetition and replace “Among these 10 tissues” by “Among them”.

- L294-296: there’s no need to repeat the n numbers in the legend (part B). Just write “same tissues and n numbers as in part A”.

- L297: in the legend, replace “form” by “from”.

- L329: in the caption, replace “…tolerance of rice” by “…tolerance to rice”.

- L340: a “s” is missing. Replace “nucleotide” by “nucleotides”.

- L342: replace “positions” by “position” since there’s only one.

- L344: a “s” is missing. Replace “gene’ expression” by “gene’s expression”.

- L351: the final dot in the legend is missing.

- L370: replace “…considered a key driver…” by “…considered as a key driver…”.

- L378: replace “specie’s” by “species’”.

- L379: replace “…ARR-B gene had nine…” by “…ARR-B genes had nine…”.

- L431: replace “italic” by “italica”.

- L466: there’s no reference for the equation used in this calculation. This should be added.

Comments on the Quality of English Language

English writting is OK. Only few errors easy to correct and listed in my comments.

Round 2

Reviewer 1 Report

Comments and Suggestions for Authors

I have reviewed the revised version of the manuscript and would like to inform you that all of my previous suggestions and comments have been addressed satisfactorily. 

The improvements made are commendable, and the revisions have significantly enhanced the clarity and quality of the paper.

Author Response

Comments 1: I have reviewed the revised version of the manuscript and would like to inform you that all of my previous suggestions and comments have been addressed satisfactorily.

Response 1: We are truly grateful to this reviewer for his/her efforts in improving our manuscript tremendously.

Comments 2: The improvements made are commendable, and the revisions have significantly enhanced the clarity and quality of the paper.

Response 2: Thank you very much.

Reviewer 2 Report

Comments and Suggestions for Authors

All the comments and problems raised were taken into account and the authors respond to all of them. Only few minor corrections remain.

Below is the list of these minor points that have to be corrected:

- L50-51: replace ”the CK receptors histidine kinases (HKs), the histidine-containing phosphotransfer (HPts)” by ”the CK receptors histidine-aspartate kinases (HKs), the histidine-containing phosphotransfer (HPts) proteins”.

- L57: replace “basing” by “based”.

- L72: since there’s a lot of data about RR in Poaceae, the assertion should be moderated. Therefore, replace “largely unexplored” by “not totally explored”.

- L102: genes have no pI but proteins do. Replace “...of the 60 RRB genes …” by “…of the 60 RRB proteins…”.

- L111: since it is known that RRBs are involved in numerous physiological and stress responses, this sentence should be less speculative. Therefore, replace “… indicating that they may also have distinct biological functions” by “…reflecting the numerous biological functions they involved in”.

- L113: a coma is missing after “pero”.

- L135: the acronym “JTT” should be explained.

- L152-153: this last sentence can be deleted since the 60 RRB genes were identified on the basis of the presence of this domain.

- L160: the word “genes” is missing after “RRB”. Replace “…most RRBs within…” by “…most RRB genes within…”.

- L161: same.

- L163: same.

- Fig. 2: in Fig 2B and C, the scale starts from 5’ to 3’ but the molecules represented are proteins, not genes, therefore the correct extremity names are Nter and Cter. Moreover, the unit should be added (aa), as it is indicated in Fig 2D.

- L245: the “o” is missing in the word  “Paceae”.

- L251: replace “four” by “five”.

- L256: same.

- Table 3: the meaning of “NAN” in the last column is not explained.

- L269: replace the caption by “Identification of cis-acting elements in OsRRB promoters”.

- L 378-379: the same sentence is repeated twice.

- L380: “RRBs” correspond to proteins and should be written in regular, not in italic.

- L381: the space between “genes” and the ref is missing.

- L384: same as L72.

- L403-412: there is no need to list the results again in this discussion section. To be deleted.

- L505: same as L380.

- L554: replace “form” by “from”.

- L641-642 : the author’s first name was used, instead of family name. The correct reference citation is Mira-Rodado, V.  The location indicated after the journal name is useless: delete this information “(Basel)”.
